# Critical calls: Circadian and seasonal periodicity in vocal activity in a breeding colony of Panamanian golden frogs (*Atelopus zeteki*)

**Alan Zigler** [1] *, **Stephanie Straw** [2], **Isao Tokuda** [3], **Ellen Bronson** [1], **Tobias Riede** [4]

**1** The Maryland Zoo in Baltimore, Baltimore, Maryland, United States of America, **2** College of Veterinary Medicine, Midwestern University, Glendale, Arizona, United States of America, **3** Graduate School of Science and Engineering, Ritsumeikan University, Kusatsu, Shiga, Japan, **4** Department of Physiology, Midwestern University, Glendale, Arizona, United States of America

* Alan.Zigler@MarylandZoo.org

## Abstract

The Panamanian golden frog (*Atelopus zeteki*) is a critically endangered species and currently is believed to survive and reproduce only in human care. Panamanian golden frog males are considerably vocal which may be an important component in their successful reproduction, though little is currently known about their calls. To better understand the behavior and vocal patterns of this species and to improve breeding efforts in the assurance colony, we employed individual sound recording of male advertisement calls and acoustic monitoring of a breeding colony to investigate variation in the vocal behavior of Panamanian golden frogs. The goal was to capture variability within and among frogs as well as patterns of periodicity over time. First, the advertisement calls from individual male Panamanian golden frogs were recorded, and acoustic parameters were analyzed for individual differences. Results suggest that male advertisement calls demonstrate individual- and population-specificity. Second, data collected through a year-long acoustic monitoring of the breeding colony were investigated for circadian and circannual periodicity. Male vocal activity revealed a circadian periodicity entrained by the daily light schedule. Seasonal periodicity was also found with highest vocal activities between December and March. The finding of a seasonal periodicity is worth noting given that the population had been bred for 20 years under constant environmental conditions. Finally, results suggest that vocal activity was responsive to daily animal care activity. Vocal activity decreased substantially when personnel entered the room and engaged in animal husbandry activities. The findings illustrate the usefulness of acoustic monitoring to provide insight into animal behavior in a zoo setting in a key breeding colony of endangered animals, and calling pattern observations may be utilized to modify husbandry practices to improve Panamanian golden frog breeding success and general care.

**Data Availability Statement:** Acoustic recordings of 13 individual frogs and one example 24-hour recording have been published on DRYAD (https://doi.org/10.5061/dryad.k98sf7mbx).

**Funding:** The author(s) received no specific funding for this work.

## Introduction

The Panamanian golden frog (*Atelopus zeteki*) (PGF) is critically endangered [1] and has not been sighted in the wild since 2009 (e.g., [2–5]). Its survival is dependent on successful breeding in human care, which faces several challenges such as health risks from exceptionally long periods of amplexus prior to oviposition [6–8]. Since limited information about the species' behavior adds to those challenges, this study was conducted to explore one component of behavior that likely accounts for a significant amount of its energy expenditure: Anuran male advertisement calling. Calling rates are positively correlated with metabolic rate and costs for calling are estimated to be 10 to 25 times greater than the resting metabolic rate [9, 10]. Here we have investigated variability among males and periodicity over time in male advertisement calls.

Male PGFs perform a combination of visual and vocal signaling: they wave their front limbs (also known as foot flagging or semaphoring) at females [11] likely as an adaptation to a noisy natural habitat near streams [12], and produce calls during the breeding season [13–15]. Research in diverse anuran species has shown that vocal behavior serves important functions in reproductive biology and provides insight into an animal's hormonal state and expectations of further reproductive behavior. Male *Xenopus laevis* and *Lithobates pipiens* vocal activity is correlated with its plasma testosterone levels [16, 17]. *Dryophytes cinerus* vocal behavior ceases after removal of gonads and re-emerges with testosterone replacement [18]. Regarding hormonal response to calls, testosterone levels in males increase after hearing advertisement calls in *Rana esculenta* [19], *Rana sphenocephala* [20], and *Dryophytes cinerus* [21], and estradiol levels in females increase after hearing male advertisement calls in *Alytes muletensis* [22] and *Engystomops pustulosus* [23]. A female's phonotaxis is also triggered by calls in *Engystomops pustulosus* and *Dryophytes chrysoscelis*, which become more likely to engage in copulatory behavior (e.g. [24–26]. Whether other components of female behavior, such as oviposition, are affected, is debated [27].

Three call types have been described for *Atelopus*: spontaneously produced male advertisement calls (aka pulse calls or long calls), release calls produced by a male typically being amplexed by another male (aka short calls), and pure tone calls [13, 14]. In general, anuran male advertisement calls demonstrate acoustic characteristics which make them both individual and species-specific, and in some cases population-specific (e.g. [28–30]). In previous investigations of *Atelopus*, only small call sample sizes were available [13, 14], therefore one goal of this study was to quantify natural acoustic variation focusing on male advertisement calls.

*Atelopus* species are vocally active during the day [15, 31–33]. In general, anuran vocal behavior is a rhythmic behavior and demonstrates periodicity over a 24-hour and seasonal period [15, 33]. The pacemakers for the observed rhythmicity are largely unknown for frogs but vocal activity often correlates with light, temperature, and humidity patterns [34–40]. Vocal activity is also affected by background noise levels [41]. Here we studied the breeding colony in the Maryland Zoo in Baltimore (MZiB) which experiences only minimal changes in light-dark phases and temperature conditions throughout the day and year, and daily care follows a stable schedule.

We began this study by collecting sound recordings from individual frogs to inform our understanding of PGF's vocal signal variability. We then recorded room acoustics over 24 hours on three days each month for 13 months between December 2019 and December 2020 to determine whether diurnal and/or circannual periodicity exists. Acoustic monitoring, or the passive acoustic surveillance of an area, territory, room, or enclosure in order to investigate the behavior of one or several species living in that space [42], has proven useful to evaluate individual or group activity patterns in diverse species (e.g. [43–46]), including frogs [47, 48]

and even to re-discover species [49]. The long-term acoustic monitoring data was analyzed for daily and seasonal periodicity. Since the MZiB population receives few environmental cues which could entrain vocal activity, seasonal changes were expected to be less prominent. Furthermore, the yearlong nature of our monitoring allowed us to see differences in call frequencies on holidays where personnel care time was halved, allowing us insight into the relationship between animal care practices and spontaneous vocal activity in an anuran species.

## Methods

### Animals and daily animal care practices

The Maryland Zoo in Baltimore maintains a breeding colony of PGFs which was established in 2001 with the importation over three years of 19 males, 19 females, and 12 juveniles of unknown gender [50]. The breeding colony includes two morphotypes (hereafter A-population and S-population) which differ in size and skin coloration. The A-population, collected from dry forest streams in a lowland location, is most distinguishable from the S-population by their smaller adult body size. The S-population was collected from a highland location with several waterfalls and mossy boulders and is roughly one third larger than the A-population frogs. The breeding colony is managed by a Species Survival Plan (SSP) as separate studbooks for each morphotype.

Frogs were kept in same-sex sibling groups of up to eight individuals in glass tanks (61x46x30 cm) with a maximum of 63 occupied tanks in a 27 m$^2$ room. At any given time between December 2019 and December 2020 there were between 385 and 333 frogs housed in the room (approx. 20% juveniles). Each tank contained about 5 cm of water, a 25 x 25 cm platform covered in sheet moss 8 cm above the tank floor, and halved terracotta pots and plastic pothos leaves for climbing and sheltering opportunities. The room was controlled for light (12 hours light; 0700–1900) using T8 Starcoat Ecolux bulbs (General Electric, Fairfiled, CT USA) and ZooMed RepiSun 10.0 UVB Reptile Lamps (ZooMed Laboratories Inc., San Luis Obispo, CA USA), which were replaced every six months. Temperature (22 ± 2° C) and humidity (60 ± 20%) were monitored.

Daily care routines included daily visual checks of all frogs. Adult animals received tank misting and feeding with springtails, fruit flies, or crickets every other day, and full tank water changes on the alternate day. In order to best standardize the total daily workload and time spent in the room, every day roughly half of the tanks in the room were misted and fed while the water was changed for the other half. Female adult frogs were given additional feedings outside of this rotation, fed a total of six days a week. The insects were dusted with a three-component vitamin and mineral supplement mix in equal parts by weight (NEKTON-MSA and NEKTON-Rep, NEKTON, Keltern, Germany; Rep-Cal D3, Rep-Cal Research Labs, Los Gatos, CA USA). Juvenile frogs were also housed in the room and were fed twice daily with supplement-dusted springtails or fruit flies.

Daily care routines were typically completed between 0900 and 1600. On the holidays of Thanksgiving and Christmas or on emergency weather days, care routines were modified to end at 1200. For a period of about 6 months following the onset of the COVID-19 pandemic in March 2020, adjustments to the daily care routine were required. Specifically, on several days, the daily care was only conducted in the afternoons between 1300 and 1600. While most frog audio monitoring was limited to days with regular 0900 to 1600 daily care routine, one recording session in November 2020 captured room acoustics during an altered 1300 to 1600 schedule due to the Thanksgiving holiday—personnel left early the day prior and arrived late the day of question, causing a 24-hour gap in the presence of personnel. Vocal activity in the

morning between 0900 and 1300 appeared to differ during that one day compared to the November average. To further assess this discrepancy, room acoustics were compared between normal (0900 to 1600) and altered (1300 to 1600) daily care routine schedules once in March 2021 and a second time in November 2021.

The study was reviewed and approved by the Maryland Zoo Research Committee (RAC#19001).

## Sound recordings

To investigate whether calls show individual differences, sixteen males were randomly selected for recording. Males were singly housed in a tank of identical composition and care as their home tank. During the single housing experiment, the experimental tank was flanked by tanks housing groups of females that were within potential line of sight of the males. The singly-housed males were given 24 hours to acclimate in the tank. Each male was then sound recorded for 48 hours. Sound was recorded with a condenser microphone (AKG C417 L Omnidirectional Lavalier Microphone; frequency range 20Hz-20kHz) attached to the lid of the tank. The sound signal was recorded at a sample rate of 44.1 kHz directly onto a computer using Avisoft Recorder software (Avisoft Bioacoustics, Berlin, Germany).

Of the sixteen frogs monitored, three frogs did not call during observations, leaving us with collections of 20 or more call recordings from 13 singly-housed males (6 from the A- and 7 from the S-population). For each frog, 20 calls were randomly selected and five acoustic parameters (fundamental frequency, call duration, pulse interval, first and second dominant frequency) were measured and analyzed. Acoustic analyses were performed using PRAAT sound analysis software (version 5.3.80 for Windows; www.praat.org). Fundamental frequency was measured in the middle of the call as the inverse of an average pulse duration. Call duration was measured as the interval between the first and last pulse of a call. Pulse interval was measured as the ratio between call duration and the total number of pulses. The first and second dominant frequencies were estimated from a short 50-ms segment from the middle of a call using linear predictive coding (LPC) function in PRAAT.

To investigate seasonal and diurnal vocal activity of the population, the room was acoustically monitored from December 2019 until December 2020. The chorus of toads in the room was monitored continuously for 24 hours on five days each month and data from three days of each month was used for analysis. An attempt was made to select days to record and analyze randomly but selection was somewhat constrained by personnel availability. Sound was recorded with a microphone (Sennheiser ME66 head and K6 phantom power module; super-cardioid-lobar pattern) that was installed centered in the room 220 cm above ground pointing directly downward. The sound signal was recorded at a sample rate of 44.1 kHz directly onto a computer using the above software. The software's trigger function was adjusted to high sensitivity in order to record the lowest known amplitude sounds from the frogs, and the recording clarity was high enough to confidently determine if one or multiple calls occurred within one trigger or if any overlapping calls occurred. The 24-hour recordings were analyzed in one-hour bins, i.e. vocal activity refers to the total number of calls recorded during a full hour. The number of advertisement calls per hour was determined by visual inspection of spectrograms, again using PRAAT sound analysis software.

Room recording was extended to four additional days outside of this year to fulfill three pairs of 24-hour recordings with normal followed by alternative daily care schedules: two days in March 2021, and two days in November 2021.

Acoustic recordings of 13 individual frogs and one example 24-hour recording have been published on DRYAD (https://doi.org/10.5061/dryad.k98sf7mbx).

## Statistical analysis

Individual differences were investigated with a multivariate ANOVA to determine whether calls from different individuals were significantly different. Univariate ANOVAs were used to determine which of the five call parameters were significantly different between individuals and the two populations. A stepwise discriminant function analysis (DFA) was used to predict group membership for each call. The result was a percentage documenting the average correct discrimination to (a) individuals and (b) two populations. The association between acoustic variation and body size, measured as body mass and snout-vent-length, was investigated with Pearson correlations.

The hypothesis that hourly means of vocal activity are equal and there is no rhythmicity in the data was tested using the chi square periodogram method [51]. If there is rhythmicity, however, the variability of periodic pattern in hourly means of vocal activity will be larger than the variability of nonperiodic patterns.

To test whether vocal activity between a day with regular daily care routine and a day with alternative daily care routine were different, we compared hourly vocal activity between 0900 and 1500 and between 0100 and 0700 using paired t-tests. If the presence of personnel in the frog room affects vocal activity, we would expect significant differences for the time window of 0900 to 1500 but not for the window of 0100 to 0700.

## Results

### Acoustic properties of male advertisement calls

Male advertisement calls were produced in single utterances consisting of up to 50 pulses (Fig 1). The pulses at the beginning of a call were spaced out more than those produced later in the call (Fig 1). Some males produced calls that were interrupted, i.e. one or more short gaps equivalent to 2 to 4 pulses was present somewhere within the call (for example males 3 and 4 in Fig 1).

Table 1 gives quantitative values for five acoustic variables measured in 20 calls from each of 13 individual frogs. Fundamental frequency ranged between 99 and 126 Hz in the A-population and between 89 and 113 Hz in the S-population. Pulse intervals ranged between 7.9 and 10.1 ms in the A-population and between 8.8 and 11.1 ms in the S-population. Call duration ranged between 390 and 500 ms in the A-population and between 470 and 650 ms in the S-population. The first dominant frequency ranged between 2149 and 2280 Hz in the A-population and between 1678 and 2390 Hz in the S-population. The second dominant frequency ranged between 5281 and 7900 Hz in the A-population and between 4759 and 7088 Hz in the S-population. Multivariate ANOVA of 260 calls from 13 toads revealed significant differences between calls from different individuals (Wilks' l = 0.52, F = 46.6, p < 0.001). Subsequent univariate tests showed that all variables except the second dominant frequency were significantly different (Table 2).

Parameters which differentiate individuals well are expected to show a greater inter-individual than intra-individual variability (Table 1). We measured within- ($CV_w$) and between ($CV_b$) individual coefficients of variation of five acoustic parameters (Table 1). Between-individual variation and average within-individual variation are presented in Table 2. The intra-individual variability approximated the inter-individual variability for four parameters (pulse interval, fundamental frequency, call duration, and second dominant frequency). The ratios of $CV_b/CV_w$ were slightly larger than 1.0, but for the first dominant frequency the ratio of $CV_b/CV_w$ was 3.75 suggesting best capacity to differentiate individual frogs (Table 2).

We used a stepwise forward DFA to investigate whether call parameters contributed to discrimination of individuals. Stepwise DFA revealed that the average correct assignment of the

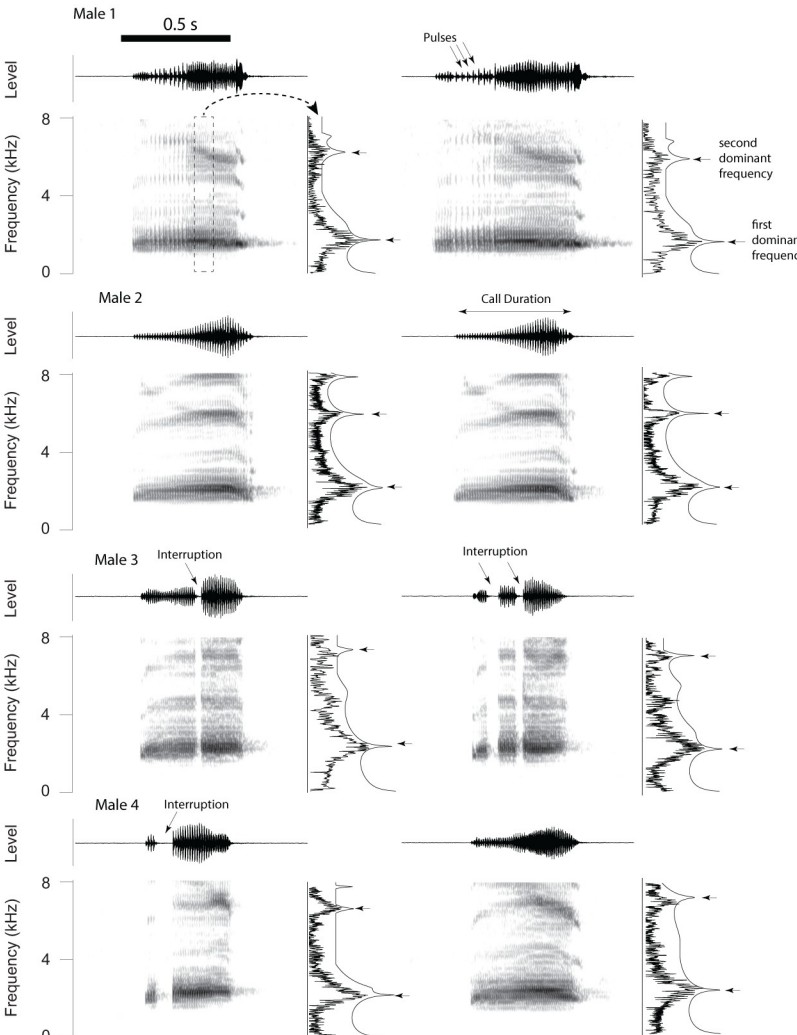

**Fig 1. Two representative advertisement calls from four male Panamanian golden frogs showing qualitative differences in pulse patterns and dominant frequency distribution.** The first few pulses of a call are spaced out more than the remaining pulses and can therefore be identified in the spectrogram (arrows). Each call sound is depicted as a time waveform (top panel) and spectrogram with an accompanying spectrum (lower panel). "Level" indicates the relative change in output voltage of microphone signal. The spectrogram illustrates the spectral content over time. The adjacent spectrum was calculated over a 50-ms segment from the middle of each call and overlaid with the LPC function.

original data set was 70.8%, i.e. on average 70.8% of the calls were correctly classified to 13 individuals. We also used a stepwise forward DFA to identify the call variables that contributed most to discrimination of the A- and S-population. Stepwise DFA revealed that the average correct assignment was 81.5% to two populations. Fig 2 is a plot of advertisement calls of individual males in space defined by the first two discriminant functions (or canonical scores). There was overlap among individual males, which explains the low $CV_w/CV_b$ ratio for four of the five parameters. The two populations separate well except for two individuals from the S-population which clustered entirely with the A-population.

Next, we investigated whether the acoustic variation was associated with body size. Body mass and snout-vent-length are reported in Table 1. Both variables were significantly associated when all 13 individuals were lumped (Pearson correlation; $r = 0.958$; $p < 0.001$) but once

**Table 1. Raw data (mean ± stdev) for 13 individual Panamanian golden frogs belonging to two populations (A-, and S-population).**

| Frog ID | BM(g)/SVL (mm)/Age (mnth) | Average F0 (Hz) | $CV_w$ F0 | Pulse interval (ms) | $CV_w$ Pulse interval | Call Duration (s) | $CV_w$ Call Duration | 1st dom Freq (Hz) | $CV_w$ 1st dom Freq | 2nd dom Freq (Hz) | $C_w$ 2nd dom Freq |
|---|---|---|---|---|---|---|---|---|---|---|---|
| **8111S** | 7.7/44.75/49 | 113.59 ± 1.76 | 1.55 | 8.81 ± 0.14 | 1.56 | 0.52± 0.04 | 7.69 | 1677.60 ±45.64 | 2.72 | 6082±316 | 5.20 |
| **8131S** | 7.5/43.51/49 | 91.22 ± 1.32 | 1.45 | 10.97± 0.16 | 1.43 | 0.47±0.02 | 4.56 | 1928.80 ±49.20 | 2.55 | 6410±2026 | 31.61 |
| **5254S** | 5.7/43.2/167 | 109.65 ± 14.90 | 13.59 | 9.25±1.01 | 10.96 | 0.49±0.03 | 5.74 | 2276.40 ±79.03 | 3.47 | 6830±784 | 11.48 |
| **7767S** | 8.0/44.75/72 | 96.06 ±7.5 | 7.81 | 10.46±0.67 | 6.42 | 0.54±0.08 | 14.28 | 2392.10 ±64.75 | 2.71 | 7088±133 | 1.88 |
| **7772S** | 7.7/42.46/72 | 94.29 ± 3.21 | 3.41 | 10.62±0.39 | 3.71 | 0.65±0.04 | 6.08 | 1867.60 ±98.54 | 5.28 | 6277±717 | 11.42 |
| **7885S** | 6.4/41.94/61 | 96.79 ± 2.49 | 2.57 | 10.34±0.26 | 2.56 | 0.51±0.04 | 7.02 | 1808.90 ±71.37 | 3.95 | 6779±622 | 9.18 |
| **6299S** | 9.0/44.19/119 | 89.98 ± 3.81 | 4.23 | 11.13±0.53 | 4.79 | 0.51±0.09 | 16.75 | 1678.20 ±57.34 | 3.42 | 4759±991 | 20.81 |
| **S-population Average:** | 7.4±1.1/ 43.5±1.1/ 84.1±43.5 | 98.8± 9.2 | 4.94 ± 4.3 | 11.0±1.9 | 4.49 ± 3.4 | 0.52±0.06 | 8.87 ± 4.6 | 1948 ±285.1 | 3.44 ± 0.9 | 6319±772 | 13.08 ± 10.1 |
| **8552A** | 3.1/32.54/34 | 121.00 ± 4.95 | 4.09 | 8.28±0.39 | 4.66 | 0.41±0.04 | 9.85 | 2257.40 ±99.07 | 4.39 | 7945.40 ±1887.02 | 23.75 |
| **8562A** | 3.7/34.35/34 | 100.13 ± 5.05 | 5.04 | 10.01±0.54 | 5.40 | 0.48±0.04 | 7.75 | 2222.80 ±79.35 | 3.57 | 7004.40 ±647.43 | 9.24 |
| **8170A** | 4.4/33.35/48 | 103.98 ± 20.25 | 19.48 | 10.13±2.83 | 27.96 | 0.39±0.05 | 12.28 | 2258.10 ±40.32 | 1.79 | 7582.30 ±867.10 | 11.44 |
| **8195A** | 3.8/33.48/48 | 99.36 ± 8.73 | 8.79 | 10.14±0.89 | 8.74 | 0.50±0.09 | 18.46 | 2280.90 ±83.21 | 3.65 | 6874.30 ±458.50 | 6.67 |
| **7902A** | 4.8/37.15/63 | 126.49 ± ±7.19 | 5.68 | 7.93± 0.45 | 5.73 | 0.42±0.04 | 9.85 | 2206.30 ±77.46 | 3.51 | 4606.10 ±1011.60 | 21.96 |
| **7905A** | 4.4/36.89/63 | 111.48 ± 8.05 | 7.22 | 9.01±0.60 | 6.66 | 0.47±0.05 | 9.99 | 2149.00 ±22.39 | 1.04 | 5281.40 ±1131.00 | 21.42 |
| **A-population Average:** | 4.0±0.6/ 34.6±1.9 48.3±13.0 | 110.41± 11.3 | 8.38 ± 5.7 | 9.3±1.0 | 9.85 ± 8.9 | 0.44±0.04 | 11.36 ± 3.8 | 2229 ±47.5 | 2.99 ± 1.3 | 6548±1320 | 15.74 ± 7.5 |

For each frog, fundamental frequency (F0), pulse interval, call duration, and first and second dominant frequency (1st and 2nd dom freq) were measured in 20 calls.
Individual frogs belong to one of two phenotypes (letter notation in the ID label). $CV_w$ = coefficient of variation within males; $CV_b$ = stdev/mean

the data set was controlled for population affiliation, the relationship did not reach significance (partial correlation: r = 0.564; p = 0.056) (Fig 3A).

Snout-vent-length was significantly associated with the first dominant frequency (Pearson correlation; r = -0.57; P = 0.042) and call duration (r = 0.623; P = 0.023) but not with

**Table 2. Mean within-male ($CV_w$) and between-male ($CV_b$) coefficients of variation, the $CV_b$/$CV_w$ ratio and the results of model II ANOVAs examining between-male variability (13 males; 20 calls per male) in Panamanian golden frogs.**

| Call property | Mean within male $CV_w$ | Between male $CV_b$ | $CV_b$/$CV_w$ | $F_{12,247}$ Between individuals | sign | $F_{2,258}$ Between A- and S-popultion | sign |
|---|---|---|---|---|---|---|---|
| Pulse Interval | 7.0±6.8 | 10.4 | 1.48 | 22.9 | P<0.001 | 36.4 | P<0.001 |
| F0 | 6.5±5.1 | 11.0 | 1.69 | 35.4 | P<0.001 | 52.1 | P<0.001 |
| Duration | 10.0±4.3 | 13.7 | 1.37 | 32.2 | P<0.001 | 92.7 | P<0.001 |
| 1st dom Freq | 3.2±1.1 | 12.0 | 3.75 | 251.6 | P<0.001 | 119.4 | P<0.001 |
| 2nd dom Freq | 14.3±8.7 | 15.9 | 1.11 | 19.3 | P<0.001 | 2.2 | P = 0.135 |

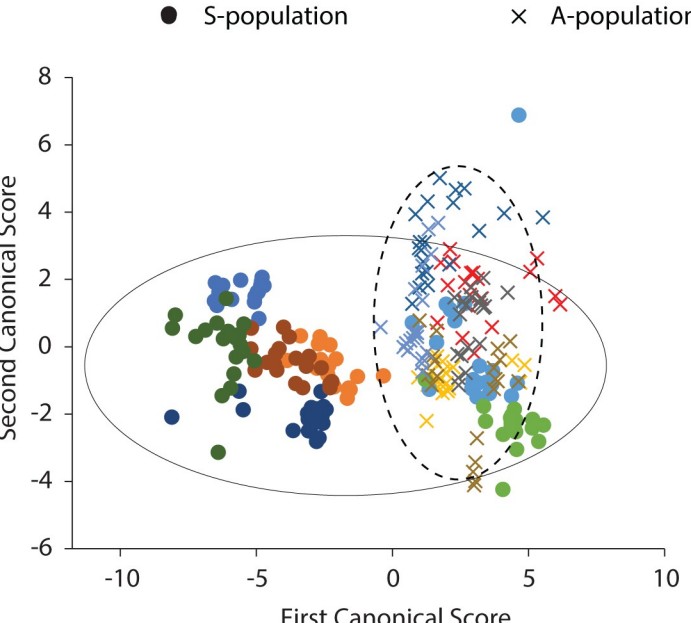

**Fig 2. Plot of the advertisement calls of 13 individual male Panamanian golden frogs in a two-dimensional signal space defined by the first two canonical scores with 95% confidence ellipses for each population.** The frogs belong to two populations (A and S) which differ in body size, skin coloration, and location of origin.

fundamental frequency (r = -0.413; P = 0.160), pulse intervals (r = 0.382; P = 0.198), or the second dominant frequency (r = -0.344; P = 0.250) (Fig 3B–3F).

## Circadian and circannual patterns of vocal activity

Fig 4A shows hourly means of vocal activity for each month between December 2019 and December 2020. Highest daily vocal activity occurred during the morning between 0700 and

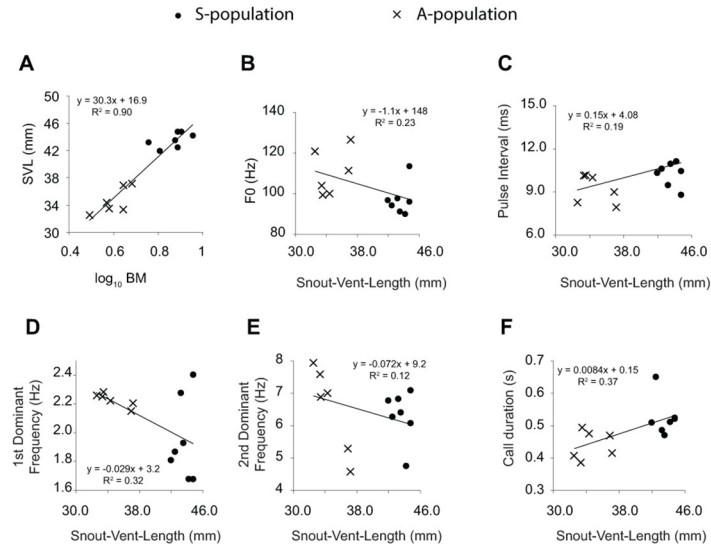

**Fig 3. Relationship between snout-vent-length and body mass (A), fundamental frequency (F0) (B), pulse interval (C), first (D) and second dominant frequency (E), and call duration (F).** The lines represent linear regression lines, with equations and $r^2$ given in each case.

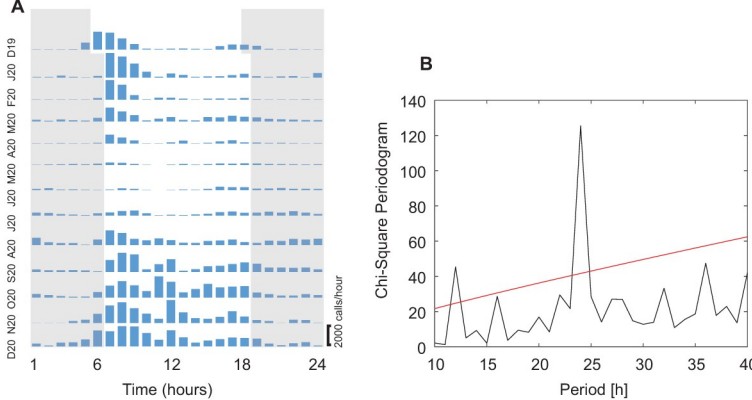

**Fig 4. A**: Vocal activity (calls/hour) over a 13 month period between December 2019 and December 2020. Grey area indicates the 12 h dark phase. **B**: Periodogram analysis of the 13 month vocal activity. The red line indicates the critical value of 0.01 level. As expected, a 24 h period was significant. A 12 h period was also significant, although less prominent. The 12 h period depicts two peaks of high vocal activity, one between 0600 and 1200 and a second after 1500.

0900 (Fig 4A). A second peak appears to exist in the late afternoon between 1600 and 1800. Between August and December 2020 there was a third peak around 1200. The periodogram identified two periods: the dominant cycle is a daily or 24-hour period and a second less dominant cycle was identified at about 12 hours and reflects the morning and afternoon peak elevated vocal activity (Fig 4B). A third cycle was not identified.

Vocal activity varied throughout the 13 months of recording. Averaging vocal activity between 0700 and 0900 indicates highest activity between December 2019 and March 2020 followed by a lower activity between April and August. Between September 2020 and December 2020 the activity increased again (Fig 5A). The 4-month average vocal activity (from 0700 until 0900) between December 2019 and March 2020 was 1065 calls/hour (stdev: 190). The 4-month average between April 2020 and July 2020 was 228 calls/hour (stdev: 178) which is more than 4 standard deviations lower than the 4-month average between December 2019 and March 2020. The contrast was even stronger when the single month with highest activity (January 2020: 1468 ±167 calls/hour) was compared to the month with the lowest activity (June 2020: 59 ±57 calls/hour), a difference of more than 8 standard deviations.

Fig 5B and 5C lists the daily average and daily extremes for temperature and humidity in the frog rooms at MZiB. For comparison, monthly averages are listed for a weather station in Panama which is located near the native origin sites (Fig 5B and 5C).

## Animal care practices and vocal activity

Vocal activity was very high during the hours after the light was turned on at 0700 (Figs 4 and 5). Typically starting around 0900, personnel enter the room to provide daily care, which lasts until about 1500 and is only shortly interrupted between 1200 and 1300 for a lunch break. The start of human activities is associated with an abrupt reduction in vocal activity. Fig 6A shows an example spectrogram of a 1-hour long monitoring period, 15 minutes before and 45 minutes after personnel has entered the room. Calls from at least 3 different animals can be identified before human activity begins at about 1015. After 1015 the recording is only triggered by human activity and no more frog calls were recorded.

The effect of personnel presence was investigated by comparing vocal activity during a regular and an alternative daily care day. The hourly vocal activity for a regular daily care routine

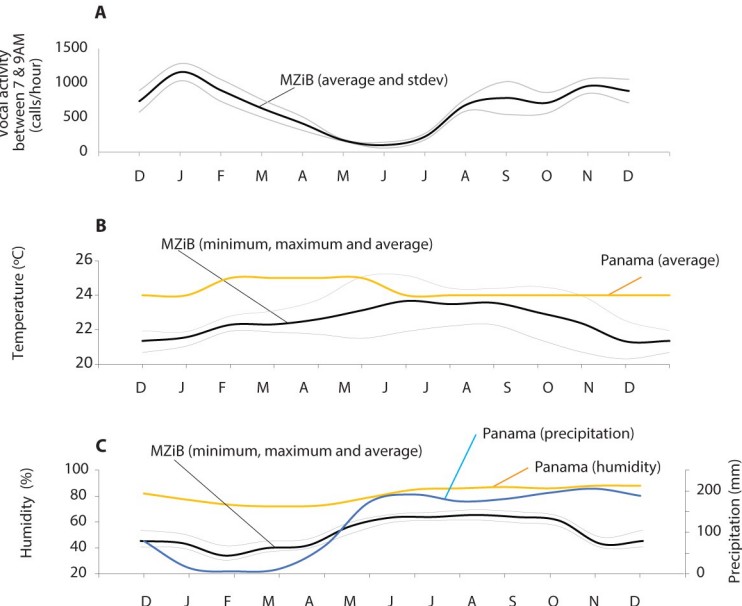

**Fig 5. Vocal activity of a Panamanian golden frog colony and climate data.** Vocal activity was measured in the colony at Maryland Zoo in Baltimore (MZIB) and averaged over two hours (**A**). Temperatures (**B**) and humidity (**C**) were available at MZIB and from weather stations near the natural habitat in Panama.

(husbandry between 0900 and 1500) and an alternative schedule (husbandry between 1300 and 1600) in the months of November 2020, May 2021 and November 2021 are shown in Fig 6B. It appears that there were at least 5 hours of increased vocal activity between 0900 and 1500 when no personnel entered the room. Hourly vocal activity during the 6 hours between 0900 and 1500 was greater during the alternative schedule than during a normal schedule day (3 months and 7 hours per day; paired t-test, df = 20, t = -7.06, p<0.001). Vocal activity between 0100 and 0700 was not different between a normal and alternate schedule day (t = -0.41, p = 0.69, df = 20) (Table 3; Fig 6B).

During a regular daily schedule all animal care is provided between 0900 and 1500. During an alternative schedule day, animal care activities take place between 1300 and 1600. No personnel enter the room between 0100 and 0700 no matter what schedule is followed. November is considered part of the breeding season of PGF, and May is outside of the breeding season.

## Discussion

Data presented here provide four new findings which inform our understanding of PGF vocal communication. First, advertisement call structure resembles the pulsed character of many toad species [52] and demonstrates individual and population specificity. Second, vocal activity revealed a circadian periodicity with highest vocal activity synchronized to time windows following the light activation and preceding light deactivation, although the presence of personnel may affect the observed pattern. Third, although the study population was not purposely subjected to annual variation in the environment, the vocal activity varied considerably between months with call frequency highest between December and March. Fourth, a response to the daily animal care activity was identified: there was a significant decrease in vocal activity between 0900 and 1500 when personnel were in the room which did not occur on adjusted days where personnel was not present.

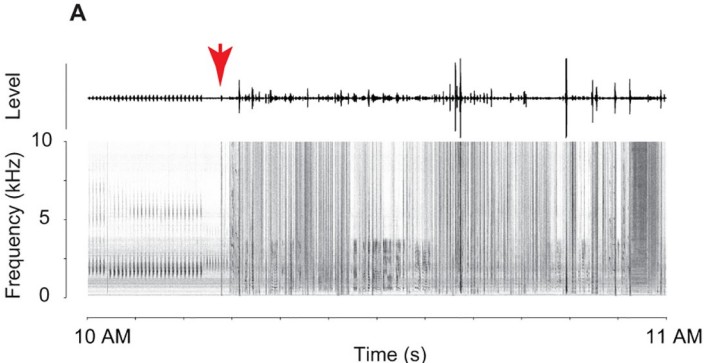

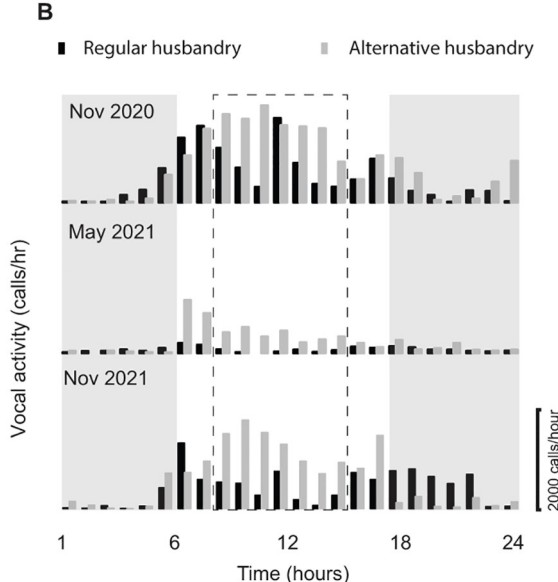

**Fig 6. Human presence reduces vocal activity. A**: Time waveform (top panel, level, relative change in output voltage of microphone signal) and spectrogram (bottom panel) of one hour of sound recording. Note that at about 1015 (red arrow), personnel enter the room and begin daily routines. The regular vertical patterns before 1015 depict frog calls. The broadband patterns after 1015 are generated by human activity. **B**: Actograms of average vocal activity during the months of November 2020, May 2021 and November 2021 with regular personnel activity between about 0900 and 1200 and between 1300 and 1600 (black). Dashed line box indicates regular personnel activity in the frog room. The paired actograms (grey) show vocal activity during a single 24 hour period in their respective months when personnel schedule was altered to 2 hours between 1400 and 1600. Note that vocal activity remained very high between 0600 and about 1400.

**Table 3. Average vocal activity (calls/hour) (mean and standard deviation averaged over 6 hours) between 0900 and 1500 and between 0100 and 0700 on regular and alternative days.**

|  | 0900–1500 | | 0100–0700 | |
|---|---|---|---|---|
|  | **Regular** | **Alternative** | **Regular** | **Alternative** |
| Nov 2020 | 635±431 | 1303±303 | 248±315 | 244±411 |
| May 2021 | 30±23 | 306±90 | 62±52 | 145±333 |
| Nov 2021 | 315±212 | 1055±330 | 231±412 | 211±279 |

The findings demonstrate that acoustic analysis of individual recordings and acoustic monitoring of a breeding colony can provide relevant insight into anuran behavior in a zoo setting. We will discuss the findings in light of this as well as potential implications for animal husbandry in an anuran breeding colony.

### Individual and population variability of male advertisement calls

*Atelopus* male advertisement calls are species-specific [13, 53]. Here we found that the first dominant frequency was less variable within individuals than between individuals. Pulse rate, call duration and fundamental frequency contributed also to the discrimination. Based on a combination of acoustic parameters measured in advertisement calls, individual PGF males could be discriminated with an above chance probability. The results resemble those reported in other toads [54] and in frogs [55].

The interpretation of the association between body size and acoustic features remains challenging. The two morphotypes are non-overlapping in body size, and slopes of regression correlations within each morphotype are different from each other. For sufficient statistical power, future investigations should consider larger sample sizes for each morphotype. Current data suggest that dominant frequency was related to body size for most PGF individuals which is in line with previous reports for other anurans [56]. However, the findings for two S-population frogs did not follow this size dependence. The reasons that those two individuals of the S-population overlapped with the A-population are not clear and future studies may benefit from an expanded collection of S-population recordings. It is tempting to speculate that their vocal differences were related to their age or health. One of the two frogs was 14 years old, the oldest frog individually recorded by four years and considerably above the average age at recording, which was roughly five years old (Table 1). The other frog was noted by personnel to have developed severe mobility issues first requiring medical intervention about one month after the calls were recorded. Vocal changes related to age or health have not been reported in Anura but old age is associated with remodeling of connective tissue in frogs such as *Microhyla ornata* [57] and *Rana muscosa* [58]. This remodeling may affect the larynx and/or the vocal tract filter.

### Diurnal periodicity in vocal activity

Anurans are vocally active either predominantly during night (e.g. [59, 60]) or during the day (e.g. [37, 61]). In the absence of predatory pressure, however, some species show elevated vocal activity when they would otherwise remain relatively quiet [62]. The PGF colony at MZiB has remained predominantly active during the day with some activity during the dark phase between August and March. In the absence of comparative data from the wild, it remains speculative whether nightly activity is a consequence of the 20 years in a predator-free environment or just a normal part of the species' repertoire.

We observed a diurnal pattern in vocal activity with two peak periods, one peak period following the onset of the light period and the second in the afternoon. Vocal activity in other *Atelopus* species has also been characterized as diurnal [31] which is often considered rare among anurans [33]. A bimodal pattern (high vocal activity at dawn and dusk) was observed with the PGF at MZiB, although there is some evidence from observing alternate care schedules that if no personnel were present through the day, we may observe a unimodal pattern (highest vocal activity midday). Both patterns are known of numerous species which are active during the day [37, 48, 63, 64].

Light plays an important role in the entrainment of vocal activity in frogs (e.g. [65]), which demands light schedules to be considered when assessing anuran welfare as it does with other captive populations such as for rodents [66] or songbirds [67]. The light regiment at MZiB

follows a constant 12-hour light phase resembling conditions in the natural Panamanian habitat (the sun rises at about 0600 and sets at about 1800 year-round). Our data suggest that vocal activity increases prior to the light onset especially between November and March (Fig 4). This behavior schedule is reflective of calling PGF in the wild, which have been observed starting calls in their suspected breeding season at roughly 0500, prior to local astronomical twilight (E. Griffith, personal communication, September 29, 2022). All lights turn on at 0700. There are reports that a sudden onset and offset of light such as the schedule maintained in the study room may have negative effects on vocal activity—for example, Hall [68] observed that experimentally introduced acute light input to natural anuran communities in ponds and streams caused the number of frogs calling to decline. The fact that calling begins prior to sudden light activation and continues steadily afterwards suggests that PGF may have an internal clock which anticipates when the lights will activate, desensitized to the sudden onset, and adjusted their behavior accordingly. Changing light activation and deactivation to a different time and observing any changes in call patterns would help to expand on this hypothesis.

Another aspect that remains unclear is the effect of human presence on the diurnal periodicity and the bimodal pattern. The start of daily care usually coincided with the morning peak of vocal activity and was associated with a decrease in vocal activity. Whether such interference during the day affected vocal activity during other times of the day seems less likely. We found no difference in vocal activity between a normal and alternative day during night time vocal activity. The effect of human activity is further discussed below.

## Seasonal periodicity

It is remarkable that PGF at MZiB continue to demonstrate seasonal periodicity despite relatively constant housing conditions which were present over the last 20 years. This suggests that a robust endogenous regulator is in place which triggers seasonal changes in sexual behavior, and also supports suggestions from other studies that calling behavior is strongly evolutionarily conserved [33]. Seasonal changes in vocal activity are well-known in numerous anuran species that were recorded in their natural habitat (e.g. [36, 60, 64, 69]). Authors often noted a correlation of high vocal activity with certain weather conditions implying that those conditions act as behavior-triggering factors. However, results from this study suggest that even in a comparatively constant environment, seasonal patterning of a spontaneous behavior remains intact over many generations.

Endogenous and environmental factors (also known as zeitgeber) entrain the behavior of vertebrates to the 24-hour or 12-month period [70]. Melatonin is one of the important time-keeping hormones in vertebrates [71] and its production is sensitive to light [72]. A high melatonin plasma concentration during night/in the absence of light suppresses vocal activity in diurnal species, for example songbirds [73], but stimulates vocal activity in nocturnal fish species [74]. Melatonin demonstrates species-specific effects in governing seasonal reproduction [75]. Long-day, spring-breeding mammals, for example, respond to increasing day length with an activation of the hypothalamus–pituitary–gonad axis as a mechanism directly regulated by photoperiod-induced changes in melatonin rhythms [75]. It is suggested that the seasonal pattern of melatonin rhythms is conserved, but animals with different temporal breeding patterns utilize the melatonin signal differently [75]. Melatonin likely also plays a role in regulating seasonal changes in sexual behavior of frogs [38, 76] but overall, to our knowledge, the understanding of endogenous and environmental factors that regulate periodicity in spontaneous behavior in anurans is limited [77].

An endogenous basis for seasonal activity helps coordinate reproductive behavior between sexes. Wells [78] suggested a dichotomous distribution along a continuum from explosive to

prolonged breeders among anurans. Explosive breeders demonstrate synchronous arrival of females to breeding sites while the female of prolonged breeders demonstrate an asynchronous arrival during a prolonged breeding season [78]. Investigations in diverse anuran groups suggest the existence of complex breeding strategies, some depending closely and others being only loosely linked to local weather patterns [79]. PGF vocal activity was elevated for four to five months between November and March. Whether *Atelopus zeteki* belongs to either of the two extremes of breeding types is unclear. Rocha-Usuga et al. [80] compared reproductive efforts and breeding behavior of a related *Atelopus* species, *Atelopus laetissumus*, during two breeding seasons. The year with average (normal) humidity and precipitation was characterized by a short breeding season. Females spawned quickly, amplexus was short and male activity was reduced. In the drier year with below average precipitation and humidity, the breeding season lasted much longer. The authors suggest that the finding supports a certain plasticity in breeding behavior and duration of breeding season.

The maintenance of circannual rhythms under constant environmental conditions have been observed in insects, birds and mammals (beetles: [81–83]; birds: [84]; mammals: [85]). Those observations make a genetic regulation likely but so far such a mechanism has remained elusive. For example, studies in beetles demonstrated a period of 37 to 40 weeks if not entrained to an environmental factor [82]. However, if an external period of 52 weeks was installed by a yearly change in the photoperiod, the beetle's rhythm was entrained to 52 weeks [83]. The free running period of the beetle's internal clock was slightly different from the 52 weeks but this period can be adjusted by an environmental zeitgeber. Our data indicate that there is an endogenous genetic mechanism to regulate the circannual rhythm in PGF, as they have maintained an annual periodicity in the absence of a known environmental zeitgeber. It remains unclear how strongly the PGF is influenced by the environment in this zoo setting. Photoperiod, which is considered to be the strongest zeitgeber in chronobiology [86], was kept constant.

While attempts were made to maintain the PGF colony room at constant temperature and humidity year round, it is worth noting that there were still slight seasonal variations in the room: humidity and temperature were lower during the winter than the rest of the year. This variation is not parallel to conditions expected in PGF native habitats, which reportedly maintain no seasonal temperature variations and a relative humidity of roughly 100 percent year-round [50]. It is also worth noting that environmental data in the colony room was collected from outside of the PGF tanks, within which likely high humidity is maintained year-round for a significant amount of time after tanks were serviced. Future studies would benefit from measurements of temperature and humidity inside of the PGF tanks for a better understanding of any seasonal variation.

### Effect of human activity on spontaneous behavior

A range of internal and external factors influence male PGF vocal activity. Data presented here suggest that human presence together with increased anthropogenic noise levels caused by required daily animal care was associated with reduced vocal activity. The effect was observed within (November 2020 and 2021) and outside (May) the core breeding season. PGF male advertisement calls are part of the species' vocal communication, and like all acoustic signals, are susceptible to background noise from the environment [87]. Therefore it is not surprising that vocal activity decreased during daily care activities. The observation that vocal activity in PGF is sensitive to human activity adds to the long list of reports that animal vocal activity is affected by human activity and anthropogenic noise in birds (e.g., [88–90]), in cetaceans (e.g., [91–93]) in fish (e.g., [94, 95]) and in anurans (e.g., [41, 96–100]).

The consequences of reduced vocal activity due to personnel presence remain unclear for PGF. Whether reduced vocal activity affects breeding success as it does for example in *Rana* spp. or *Hyla* spp [19–21] is left to future investigations. Current reproductive challenges in this species [8] may be alleviated by better understanding the role of vocalizations in PGF breeding behavior and potentially adjusting husbandry practices to reduce personnel's impact on call rates if a high male vocal activity is deemed important, although this remains to be carefully explored. Considering the energetic costs of male advertisement calling [9], it seems important to include passive acoustic monitoring of a species vocal activity into the design of daily animal care schedules.

## Conclusions

The yearlong acoustic monitoring of a PGF breeding colony has delivered new insights into the species' breeding biology confirming the usefulness of the technique [47]. It is a critically important component of animal welfare to both understand a species' biological rhythms and to know how to provide opportunities for a species to express or maintain them [101, 102]. As such, the findings from this study may have relevance for husbandry and conservation efforts. Since male vocal activity is a critical cue for female physiology and behavior (e.g., [22–26]), the careful consideration of the vocal activity patterns of this species in the husbandry and reproductive planning of PGF may be a promising factor towards identifying cues and improving anuran breeding programs. The negative effect of anthropogenic noise and human presence on breeding success is well known in laboratory animals (e.g., [102, 103]) and simple adjustments to daily animal care routines, as well as environmental factors such as temperature, humidity, and light control might improve success by entraining activities to the frog's internal clock.

One challenge in captive PGF breeding is the failure of gravid females to release eggs (oviposition) [6–8]. PGFs show an unusually long and vigorous amplexus period which imposes additional costs for both males and females [15, 104, 105] and is suspected to contribute to high mortality rates during the breeding season [6–8]. Vocal activity may be important for triggering a female's hormonal cascade leading to eventual successful oviposition, an idea that is not new [27]. Future research in PGF may investigate what, if any, behavioral or hormonal responses to vocalizations exist among males and females of the species, and with our gained understanding of several acoustic features showing variability between both populations and individuals, whether modifications are associated with greater reproductive success.

## Acknowledgments

We thank the Panamanian Golden Frog care staff at MZiB for their excellent care and for flexibility regarding added responsibility from our yearlong acoustic surveillance: Kevin Barrett, Danielle Regan, Autumn Aaron, Paul Brandenburger, Katie Coiner, Marietta Cox, Robert Healy, and Christopher McIntosh. We thank MZiB staff member Katharine Mantzouris especially for her assistance in data management and for performing time-sensitive research tasks when researchers could not enter the study space during the onset of the COVID-19 pandemic.

Stephanie Straw was supported by a MWU CVM fellowship.

## Author Contributions

**Conceptualization:** Alan Zigler, Ellen Bronson, Tobias Riede.

**Data curation:** Tobias Riede.

**Formal analysis:** Stephanie Straw, Isao Tokuda, Tobias Riede.

**Funding acquisition:** Ellen Bronson, Tobias Riede.

**Investigation:** Alan Zigler.

**Methodology:** Alan Zigler, Isao Tokuda, Ellen Bronson, Tobias Riede.

**Project administration:** Alan Zigler, Ellen Bronson.

**Resources:** Ellen Bronson, Tobias Riede.

**Software:** Tobias Riede.

**Supervision:** Ellen Bronson, Tobias Riede.

**Writing – original draft:** Alan Zigler.

**Writing – review & editing:** Ellen Bronson, Tobias Riede.

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
