## [Decision Letter · Decision Letter 0]

6 Mar 2023

PONE-D-22-32235Critical calls: an investigation of Panamanian golden frog (Atelopus zeteki) vocal behavior in human carePLOS ONE

Dear Dr. Zigler,

Thank you for submitting your manuscript to PLOS ONE. After careful consideration, we feel that it has merit but does not fully meet PLOS ONE’s publication criteria as it currently stands. Therefore, we invite you to submit a revised version of the manuscript that addresses the points raised during the review process.

Due to difficulty finding suitable reviewers with time to contribute to the review process your manuscript has been delayed longer than I would have liked.  Therefore, I have read and evaluated the manuscript prior to reading the reviewer comments.  In this case, I am acting as both reviewer and editor. Prior to resubmission, please address all of the comments and concerns with corrections, additions, or explanations.   In my evaluation of the manuscript, I have three areas that I would like addressed in more detail.First, please explain in more detail your individual call recording procedure.  Group housing conditions are well described, but how individuals were transferred and housed individually is not sufficiently explained.  Please see corresponding concern from the Reviewer regarding this issue.  Include a complete description of the process required for individual housing (time given to acclimate to the new cage), dimensions, etc.  Further, a description of how individual housing vs. social housing might affect vocal behavior would be relevant - note Reviewer comment on this as well.Second, I have one concern about statistical analyses, and this centers on your bivariate correlation analyses of body mass features and acoustic features.  First, this analysis is not described in the methods.  Second, your analysis is challenging due to the physical differences between the two morphotypes, that have non-overlapping groups of data - I feel the data would be stronger if your slopes for the regression correlations within each morphotype were not significantly different from each other, however you lack sufficient statistical power to test this idea (in my estimation based on the sample sizes for each morphotype).  I would like to suggest collecting more data for this analysis, but I am sympathetic to the challenges of this experiment.  I suggest you consider alternative analysis options or give caveats for this section due to these group differences. Last issue here, when reporting a simple regression, a lower case "r" should be used.  The upper case R indicates multiple regression. Third, and last, in my view, Figure 1 would more clearly show spectral content differences in the calls if the power spectrum was displayed to the left side of the spectrogram.  This would help to clarify what you mean by the first and second dominant frequencies.  In the methods, please briefly explain how the first and second dominant frequency are determined. Please note Reviewer concerns regarding this issue as well. Please submit your revised manuscript by Apr 20 2023 11:59PM. If you will need more time than this to complete your revisions, please reply to this message or contact the journal office at plosone@plos.org. Please include the following items when submitting your revised manuscript:A rebuttal letter that responds to each point raised by the academic editor and reviewer(s). You should upload this letter as a separate file labeled 'Response to Reviewers'.A marked-up copy of your manuscript that highlights changes made to the original version. You should upload this as a separate file labeled 'Revised Manuscript with Track Changes'.An unmarked version of your revised paper without tracked changes. You should upload this as a separate file labeled 'Manuscript'.If applicable, we recommend that you deposit your laboratory protocols in protocols.io to enhance the reproducibility of your results. Protocols.io assigns your protocol its own identifier (DOI) so that it can be cited independently in the future. For instructions see: https://journals.plos.org/plosone/s/submission-guidelines#loc-laboratory-protocols. Additionally, PLOS ONE offers an option for publishing peer-reviewed Lab Protocol articles, which describe protocols hosted on protocols.io. Read more information on sharing protocols at https://plos.org/protocols?utm_medium=editorial-email&utm_source=authorletters&utm_campaign=protocols.

We look forward to receiving your revised manuscript.

Kind regards,

Brenton G. Cooper, Ph.D.

Academic Editor

PLOS ONE

“Stephanie Straw was supported by the Midwestern University College of Veterinary Medicine Summer Research Fellowship.”

Reviewers' comments:

Reviewer's Responses to Questions

**Comments to the Author**

1. Is the manuscript technically sound, and do the data support the conclusions?

Reviewer #1: Yes

2. Has the statistical analysis been performed appropriately and rigorously? 

Reviewer #1: Yes

3. Have the authors made all data underlying the findings in their manuscript fully available?

Reviewer #1: No

4. Is the manuscript presented in an intelligible fashion and written in standard English?

Reviewer #1: Yes

5. Review Comments to the Author

Reviewer #1: Manuscript entitled " Critical calls: An investigation of Panamanian golden frog (Atelopus zeteki) vocal behavior in human care" found to be a significant study. The species has not been sighted in the wild since 2009 and is believed to survive and reproduce only in human care. The study has investigated the call variations within and among the individuals in captivity and patterns of periodicity over time. Findings of the work provide relevant insight into anuran vocal behaviour in a zoo setting and provides potential implications for animal husbandry in an anuran breeding colony.

The manuscript is well written and the figures are sufficient to illustrate the findings of the study. In sum, I have no reservations against the publications of this manuscript and consider it to be a substantial increase to our knowledge on this critically endangered Atelopus zeteki.

Some of the major and minor issues I find are listed below:

Line 185 – When investigating the population’s vocal activity how you count the total number of calls hourly. The recordings might have lot of overlapping calls and many low amplitude calls. Normally toads are making choruses.

Line 227– In introduction section good to have a separate paragraph to explain the physiological aspects of calling such as metabolic rate, hormonal state, etc….

Line 223 – How did you select 20 calls from the sample recordings? And they might have periodic differences in call characters.

Line 227– Better explain briefly about the call structure and unique properties of Atelopus zeteki to give a better understanding at the beginning. For example what you mean by first and second dominant frequencies?

Line 236– Table 1, cant see the complete table

Line 252– It says CVb/CVw were larger than 1 for the first dominant frequency. But table 2 shows the value is larger than one for all variables. Reword the sentence

Line 262– Two individuals from S population means 2/7 = 29%. Increasing sample size might provide much better representation.

Line 344 – In methodology mentioned the room was acoustically monitored from December 2019 until December 2020? But in figure 6 shown actograms of 2021

• I wonder why the calling activities are comparatively higher around 1200 during regular husbandry. Does that indicate the lunch break (absence of the personnel in the frog room) ?

Line 366 – Everywhere else mentioned the alternative schedule is between 1300 and 1600 ???

Line 374 – First finding is not new. Already known information. Isn’t it?

Line 433 – Considering the human disturbances it is unfair to claim that they have two peaks. Though the disturbances happen between 1300 to 1600, figure 6B shows almost one cycle in alternative schedule day.

Fig1– In your figure male 2,3 and 4 clearly show 3-4 frequency bands. You have considered first and second dominant frequency only. And it is not clear which band you selected as second dominant frequency. Further the figure shows they modulate the frequency. So their starting frequency and end frequency is different. Crocroft (1990) has also mentioned about the frequency modulation of the species.

Fig2 – Draw and show the confidence ellipses.

•Good if you could make available some of your call recordings as supplementary materials. Consider deposit in a public repository and share the link.

6. PLOS authors have the option to publish the peer review history of their article (what does this mean?). If published, this will include your full peer review and any attached files.

Reviewer #1: **Yes: **Nayana Wijayathilaka

---

## [Author Response · Author response to Decision Letter 0]

9 May 2023

Reviewer 1

1. First, please explain in more detail your individual call recording procedure. Group housing conditions are well described, but how individuals were transferred and housed individually is not sufficiently explained. Please see corresponding concern from the Reviewer regarding this issue. Include a complete description of the process required for individual housing (time given to acclimate to the new cage), dimensions, etc. Further, a description of how individual housing vs. social housing might affect vocal behavior would be relevant - note Reviewer comment on this as well.

RESPONSE: We have added detail explaining individual housing and acoustic monitoring in the method section. 

2. Second, I have one concern about statistical analyses, and this centers on your bivariate correlation analyses of body mass features and acoustic features. First, this analysis is not described in the methods. Second, your analysis is challenging due to the physical differences between the two morphotypes, that have non-overlapping groups of data - I feel the data would be stronger if your slopes for the regression correlations within each morphotype were not significantly different from each other, however you lack sufficient statistical power to test this idea (in my estimation based on the sample sizes for each morphotype). I would like to suggest collecting more data for this analysis, but I am sympathetic to the challenges of this experiment. I suggest you consider alternative analysis options or give caveats for this section due to these group differences. Last issue here, when reporting a simple regression, a lower case "r" should be used. The upper case R indicates multiple regression. 

RESPONSE: Unfortunately, it is not possible to add more individuals to the analysis. Therefore we have revised the discussion to explain the limitations of the current analysis, and we have reduced/condensed the discussion to not over-state the relevance of the correlational results. The methods were revised to include information on the Pearson correlation analysis.

3. Third, and last, in my view, Figure 1 would more clearly show spectral content differences in the calls if the power spectrum was displayed to the left side of the spectrogram. This would help to clarify what you mean by the first and second dominant frequencies. In the methods, please briefly explain how the first and second dominant frequency are determined. Please note Reviewer concerns regarding this issue as well.

RESPONSE: Figure 1 has been revised as suggested.

Reviewer 2

1. Line 185 – When investigating the population’s vocal activity how you count the total number of calls hourly. The recordings might have lot of overlapping calls and many low amplitude calls. Normally toads are making choruses.

RESPONSE: Our process for counting calls within a day’s recordings allowed us the opportunity to catch any overlapping calls – we would see patterned single calls within a generated spectrogram and play back audio for any recordings that do not look to fit a standard call structure, which we could confidently use to confirm if the recording included overlap. To miss low amplitude calls is a valid concern, however, the vocal activity measure used here is likely to be a representation of overall population-wide activity because toads tend to make choruses. Furthermore, we took great care to confirm that frog calls originating in any tank would trigger recording. Observations during the initial month-long setup we confirmed by simultaneous observations of tanks with calling toads and the AVISOFT recorder that sensitivity was sufficient to capture calls from each tank. We have added additional detail to this section of the methodology to address this.

2. Line 227– In introduction section good to have a separate paragraph to explain the physiological aspects of calling such as metabolic rate, hormonal state, etc….

RESPONSE: We appreciate the suggestion by the reviewer. Please note that we indicate in the first paragraph of the introduction that “costs for calling are estimated to be 10 to 25 times greater than the resting metabolic rate.” Furthermore the second paragraph highlights the importance of the hormonal state (“Research in diverse anuran species has shown that vocal behavior serves important functions in reproductive biology and provides insight into an animal’s hormonal state and expectations of further reproductive behavior.”). We kindly ask to not further expand the Introduction.

3. Line 223 – How did you select 20 calls from the sample recordings? And they might have periodic differences in call characters.

RESPONSE: We have added additional detail to the method section.

We also acknowledge in the discussion section that future research should increase the individual sample size to explore the effects of season, reproductive state, age, and body size. 

4. Line 227– Better explain briefly about the call structure and unique properties of Atelopus zeteki to give a better understanding at the beginning. For example what you mean by first and second dominant frequencies?

RESPONSE: We have revised Figure 1 in order to better explain the acoustic parameters used here.

5. Line 236– Table 1, cant see the complete table

RESPONSE: We have modified the document so that Table 1 is on a separate page in landscape.

6. Line 252– It says CVb/CVw were larger than 1 for the first dominant frequency. But table 2 shows the value is larger than one for all variables. Reword the sentence

RESPONSE: The text has been revised.

7. Line 262– Two individuals from S population means 2/7 = 29%. Increasing sample size might provide much better representation.

RESPONSE: We have revised the discussion and discuss limitations of the current approach and possible routes for future studies. 

8. Line 344 – In methodology mentioned the room was acoustically monitored from December 2019 until December 2020? But in figure 6 shown actograms of 2021

RESPONSE: In order to assess the effects of human activity on vocalizations, we needed two additional pairs of comparable 24-hour periods (normal vs. altered schedule) outside of the year we recorded all other calls, which occurred in 2021. We included a paragraph in methodology to better indicate this.

9. I wonder why the calling activities are comparatively higher around 1200 during regular husbandry. Does that indicate the lunch break (absence of the personnel in the frog room)?

RESPONSE: Call activity increases between 1200-1300 when personnel leave the room for a lunch break. This observation is of critical importance and may affect how future husbandry schedules are designed.

10. Line 366 – Everywhere else mentioned the alternative schedule is between 1300 and 1600 ???

RESPONSE: We have corrected this error.

11. Line 374 – First finding is not new. Already known information. Isn’t it?

RESPONSE: We have revised this sentence. 

12. Line 433 – Considering the human disturbances it is unfair to claim that they have two peaks. Though the disturbances happen between 1300 to 1600, figure 6B shows almost one cycle in alternative schedule day.

RESPONSE: The periodogram analysis identified two cycles, 24 hours and 12 hours. Additional peaks did not reach threshold. We hope that the revisions of the discussion section sufficiently explain multiple potential sources for disturbing a natural cyclicity. Please note that figure 6B are only three recordings and do not allow conclusions about robust periods in vocal activity.

13. Fig 1– In your figure male 2,3 and 4 clearly show 3-4 frequency bands. You have considered first and second dominant frequency only. And it is not clear which band you selected as second dominant frequency. Further the figure shows they modulate the frequency. So their starting frequency and end frequency is different. Crocroft (1990) has also mentioned about the frequency modulation of the species.

RESPONSE: We have revised Figure 1 to better illustrate the parameters used here.

14. Fig 2 – Draw and show the confidence ellipses.

RESPONSE: Figure 2 has been modified to include confidence ellipses.

15. Good if you could make available some of your call recordings as supplementary materials. Consider deposit in a public repository and share the link.

RESPONSE: Data have been published on DRYAD.

Zigler, Alan; Riede, Tobias (2023), Critical Calls: Circadian and seasonal periodicity in vocal activity in a breeding colony of Panamanian golden frogs (Atelopus zeteki), Dryad, Dataset, https://doi.org/10.5061/dryad.k98sf7mbx

---

## [Editor Report · Decision Letter 1]

19 May 2023

Critical Calls: Circadian and seasonal periodicity in vocal activity in a breeding colony of Panamanian golden frogs (Atelopus zeteki)

PONE-D-22-32235R1

Dear Dr. Zigler,

We’re pleased to inform you that your manuscript has been judged scientifically suitable for publication and will be formally accepted for publication once it meets all outstanding technical requirements.

Kind regards,

Brenton G. Cooper, Ph.D.

Academic Editor

PLOS ONE
---

## [Editor Report · Acceptance letter]

30 May 2023

PONE-D-22-32235R1 

Critical Calls: Circadian and seasonal periodicity in vocal activity in a breeding colony of Panamanian golden frogs (*Atelopus zeteki*) 

Dear Dr. Zigler:

I'm pleased to inform you that your manuscript has been deemed suitable for publication in PLOS ONE. Congratulations! Your manuscript is now with our production department. 

Kind regards, 

on behalf of

Dr. Brenton G. Cooper 

Academic Editor

PLOS ONE